# Impacts of the Invasive Alien *Carpobrotus* spp. on Coastal Habitats on a Mediterranean Island (Giglio Island, Central Italy)

**DOI:** 10.3390/plants11202802

**Published:** 2022-10-21

**Authors:** Michele Mugnai, Renato Benesperi, Daniele Viciani, Giulio Ferretti, Michele Giunti, Francesca Giannini, Lorenzo Lazzaro

**Affiliations:** 1Department of Biology, University of Florence, Via La Pira 4, 50121 Firenze, Italy; 2University Museum System, Botanical Garden “Giardino dei Semplici”, University of Florence, Via Micheli 3, 50121 Firenze, Italy; 3Nature and Environment Management Operators s.r.l., Piazza M. D’Azeglio 11, 50121 Firenze, Italy; 4National Park of the Tuscan Archipelago, 57037 Portoferraio, Italy

**Keywords:** biodiversity, community ecology, conservation, endemic species, indicator species, maquis, nestedness, shrubland, turnover, vegetation

## Abstract

*Carpobrotus acinaciformis* and *C. edulis* are well-known invasive alien plants native to South Africa, whose detrimental effects on native communities are widely documented in the Mediterranean basin and thus largely managed in coastal ecosystems. Most of the literature on these species focuses on their impacts on habitats of sandy coastal dunes, while the effects of *Carpobrotus* spp. invasion on other habitats such as rocky cliffs and coastal scrubs and garrigues are almost neglected. We present a study case conducted on a small Mediterranean island where *Carpobrotus* spp. invaded three different natural habitats listed within the Habitat Directive 92/43/CEE (Natura 2000 codes 1240, 1430, and 5320). We surveyed the presence and abundance of native species and *Carpobrotus* spp. on 44 permanent square plots of 4 m^2^ in invaded and uninvaded areas in each of the three habitats. We found impacts on plant alpha diversity (intended as the species diversity within each sampled plot) in all the habitats investigated in terms of a decrease in species richness, Shannon index, and abundance. Invaded communities also showed a severe change in species composition with a strong homogenization of the floras of the three habitats. Finally, the negative effect of invasion emerged even through the analyses of beta diversity (expressing the species diversity among sampled plots of the same habitat type), with *Carpobrotus* spp. replacing a large set of native species.

## 1. Introduction

Biological invasions represent one of the most dramatic threats to biodiversity, contributing substantially to the widespread and accelerated decline in Earth’s biodiversity and associated benefits to people from nature [1,2], a situation even more dramatic considering that an increase in the rate of new introductions is foreseen in the future [3]. Invasive alien plants (IAPs) can exert several deleterious impacts on native communities, leading to a local decrease in plant and animal species richness and diversity [4,5,6]. Indeed, IAPSs can induce cascade effects linked to dramatic changes in the structure and function of invaded ecosystems [5,7,8] and resulting in the reduction of the distinctiveness of local biological communities [9]. Islands and coastal mainland regions have been recently individuated as hotspots of established alien species richness across multiple taxonomic groups [10]. In this context, the deleterious effect can be even higher, and IAPs presence and impacts are well documented on Mediterranean island ecosystems [11,12,13,14,15]. Indeed, the Mediterranean basin, with its complex system of archipelagos, islands and islets, represents an important area of species diversity. It is usually reported that Mediterranean basin vascular flora amount at about 24,000–25,000 species, accounting for 10% of world plant richness, with at least 13,000 endemics, with rates of endemism often exceeding 10%, and sometimes 20%, of local flora (see [12]).

In Europe, IAPs are considered one of the major drivers of changes in natural and semi-natural habitats and their presence increases the probability of unfavorable conservation status of natural habitats [16], being linked to a general deterioration of biodiversity and to the alteration of habitat structure and functions in plant communities [15,17]. Indeed, there is a general awareness of the potential impacts of IAPs on native plant communities and on the habitats of Community Interest listed in the Habitats Directive (Council Directive no. 92/43/EEC, hereafter N2000 habitats), even if for several habitats and species there is lack of direct evidence [18]. The importance of the N2000 network in tackling the risks posed by biological invasions was underlined by the European Commission in the EU 2020 Biodiversity Strategy, and further emphasized in the recent EU 2030 Biodiversity Strategy. However, the N2000 network efficacy in decreasing the vulnerability to invasive alien species is largely still unknown [18,19]. Accordingly, data on the presence and impact of IAS on the N2000 habitats and on the N2000 network are crucial to counter their detrimental impacts [18].

The species of the genus *Carpobrotus* N.E.Br. (Aizoaceae)—in particular *C. acinaciformis* (L.) L.Bolus and *C. edulis* (L.) N.E.Br. and their hybrids—are considered among the most abundant and most investigated IAPs in the whole Mediterranean basin, where they display a very high invasive potential [12,15,20]. The invasion by *Carpobrotus* spp. causes strong negative impacts on the ecology of invaded ecosystems, mainly sand dunes and rocky sea cliffs [20], with significant changes in the invaded ecosystems at a variety of scales [21,22,23,24]. Impacts have been well depicted on several ecosystem components and processes, from plant biodiversity and vegetation structure [13,21,25,26,27,28] to soil conditions and physico-chemical and biological processes [28,29,30,31], these latter resulting also in the reduction in seed germination and survival of seedlings of the native plants [30,32,33]. In the whole Mediterranean basin, these species have been targeted by several projects of control, mostly fostered by local stakeholders, but also often linked to the EU LIFE program (see also [20], and especially on coastal dunes in the Mediterranean basin *Carpobrotus* is the genus with the largest number of records of control actions [12,34]. In Italy *Carpobrotus* spp. are considered invasive and among the most threatening invasive plants [15], with well-documented impacts at the community level and on N2000 habitats, but with evidence mostly restricted to sand dune habitats (particularly N2000 habitats 2120 “Shifting dunes along the shoreline with *Ammophila arenaria* (white dunes)”, 2210 “*Crucianellion maritimae* fixed beach dunes” and 2250* “Coastal dunes with *Juniperus* spp.”). However, these are not the only habitat types invaded by these IAPs in Italy: in Tuscany, it is particularly invasive in the Tuscan Archipelago [35], where it is one of the most harmful invasive alien species, particularly affecting the rocky cliffs coastal vegetation of the islands of the Archipelago [14].

Hence, studies on the impacts related to IAPs are very important as they represent a valuable source of information necessary to lay the basis for any generalization on the scenario of biological invasions and are an important tool to implement and enforce more effective management strategies [2,36]. To provide a more comprehensive understanding of impacts, composite approaches are needed, assessing the effects of invasive plants on several response variables [36]. Moreover, to provide a better comprehension of the ecological process caused by IAPSs invasion, the study of beta diversity could provide important insights. Beta diversity is defined as the ratio between gamma (regional) and alpha (local) diversities, and it essentially quantifies the number of different communities in the region [37]. Indeed, beta diversity studies can provide considerable insights into the importance of deterministic and stochastic processes in generating community structure along spatial and ecological gradients [37,38], and have been already used to provide insights into drivers and mechanisms of invasion and assembly of alien communities at a broad spatial scale [39,40]. Decomposing beta diversity in different components (i.e., disentangling the contribution of spatial turnover and nestedness to beta diversity patterns) allows us to highlight patterns linked to the biological phenomena in the act and is essential for analyzing the causality of the processes underlying biodiversity [37].

Within this work, we aimed to assess the impact of *Carpobrotus* spp. on the native communities at Giglio Island (Tuscan Archipelago, Italy), providing a baseline of data on the impacts of these IAPs on invaded ecosystems as pre-intervention monitoring linked to the control of *Carpobrotus* spp. foreseen within the LIFE project LETSGO GIGLIO “Less alien species in the Tuscan Archipelago: new actions to protect Giglio island habitats” (www.lifegogiglio.eu, accessed on 16 October 2022). Moreover, we aimed at providing comprehension of ecological processes in the act to better inform conservation and restoration efforts. Accordingly, we aimed to (i) verify the impact of *Carpobrotus* spp. on native species richness, diversity, and total cover, (ii) evaluate the impacts on the native species composition of invaded communities, and (iii) assess the main processes in the act (turnover vs. species loss) focusing on beta diversity features of invaded and non-invaded communities. Toward these aims, we monitored a series of vegetation plots within the three main coastal habitats invaded by *Carpobrotus* spp. on Giglio island (N2000 habitats 1240: vegetated sea cliffs of the Mediterranean coasts with endemic *Limonium* spp., 1430: Halo-nitrophilous scrubs (*Pegano-Salsoletea*) and 5320: Low formations of *Euphorbia* close to cliffs).

## 2. Results

The sampling led to the identification of 74 species in 88 plots (see Appendix A). Vegetating cover varied substantially in Habitat 1240 between Control and Invaded plots, while it was more consistent in Habitats 1430 and 5320, even if in these habitats there was a high variation between 2020 and 2021 (see Appendix A). The results generally showed in all three habitats the presence of significant impacts on alpha diversity linked to *Carpobrotus* spp. invasion (Table 1).

*Carpobrotus* spp. cover was stable in 2020 and 2021 around 60% in habitat 1240 and varied significantly from about 50% in 2020 to 25% in 2021 in habitat 1430 and from about 75% in 2020 to 50% in 2021 in habitat 5320 (Figure 1A). Noteworthy, a few *Carpobrotus* spp. seedlings arrived in 2021 in four control plots: two in habitat 1430 and two in habitat 5320 (see zoomed bars in Figure 1A). The *Carpobrotus* spp. dead litter showed an inverse variation, being more or less stable around 20% in habitat 1240 and increasing in 2021 in both habitat 1430 and habitat 5320 (Figure 1B).

We detected highly significant and marked loss of total abundance of native species in the plots monitored in all three habitats (Figure 2A, Table 1). As to the impacts on alpha diversity, we detected a significant decrease in native plant species richness in the three habitats (Figure 2B, Table 1), with habitat 1240 and habitat 1430 showing an effect irrespective of the year of sampling, while for habitat 5320, the effect varied slightly in the two survey years, with a smaller difference in 2021.

With regard to Shannon (H’) diversity (Figure 2C, Table 1), while habitat 1240 did not show any significant difference among invaded and control plots, habitat 5320 showed a significant decrease in diversity in the invaded areas irrespective of the year of sampling. For habitat 1430, on the other hand, the effect varies in the two survey years, with a comparable diversity in 2021 between controls and invaded areas.

From the compositional point of view, both the ISA (Table 2) and the NMDS (Figure 3) testify the important changes linked to the invasion by *Carpobrotus* spp.

The ISA showed a dramatic drop in indicator species in the invaded plots compared to the controls, also testifying important compositional differences among the three habitats in their natural conditions (i.e., in the control plots). The habitat 1240 control plots had only one indicator species (*Limonium planasie*) while the other two habitats showed a very high number of indicator species in the controls (16 and 21, respectively, in habitat 1430 and habitat 5320). In the invaded status, all three habitats showed very few indicator species, with *Carpobrotus* spp. being the indicator species in only habitat 5320. The NMDS analysis (stress = 0.092, non-metric fit R^2^ = 0.991, linear fit R^2^ = 0.964) highlighted a well-defined differentiation (as expected) between the control plots of the three habitats, which lay on the upper area of the plots. Conversely, the communities appear closer when invaded by *Carpobrotus* spp. (lower part of the plot), and particularly habitat 5320 and habitat 1240 invaded plots almost overlap, while habitat 1430 invaded plots are closer to the respective control plots.

Through the assessment of beta diversity, we found different patterns for the three habitats considered. Within habitats 1240 and 5320, in both years we found that the higher levels of beta diversity are between invaded and not-invaded communities, with the greater portion due to species turnover (Figure 4).

Regarding beta diversity within habitat 1430, we found more comparable patterns of diversity within invaded, control, and between control and invaded communities: for 2020, the comparison of control–invaded results were slightly higher, while for 2021, diversity within invaded communities it was higher. In those three cases of comparison (for both 2020 and 2021), the main contribution to beta diversity of habitat 1430 is due to species turnover.

## 3. Discussion

Our results provide significant evidence on the impacts of *Carpobrotus* spp. on the rocky cliff coastal habitats of Giglio island and offer interesting insight into the ecological processes acting on the invaded sites. The very high impact of *Carpobrotus* spp. at the alpha diversity level has been frequently reported, inducing a decrease in species richness [22,24,26,41,42] of the H’ diversity index [26,41,42] and of native plants biomass [22]. However, most of the literature highlighting the general trend of native communities’ depletion when invaded by *Carpobrotus* spp. is focused on sandy dune habitats (see [15,20]) while the impacts on sea cliff communities are still rarely studied compared to the other types of coastal habitats. Our case brings evidence on the impacts exerted on rocky cliff habitats and vegetation (represented by the mosaic of habitats 1240 and 5320) and coastal cliff shrub vegetation (habitat 5320). Our results are in line with the findings of Buisson et al. [43], who found strong differences in species richness and community composition before and after its removal. We found some differences in the magnitude of the impact. Particularly to 1240, the depletion was less though compared to the other habitats considered, probably as a consequence of the pauci-specific communities typical of this habitat [44]. However, the impact registered is still noteworthy since the poor flora of such habitat includes endemic and high conservation value species such as *Limonium sommerianum* [44]. The stronger impact of *Carpobrotus* spp. on native communities for habitat 1430 and habitat 5320 is therefore connected to the normal greater specific richness (note that habitat 1430 is represented on Giglio Island by communities of the syntaxonomic alliance *Artemision arborescentis*) which characterizes them, increasing the effect of the species disappearance caused by *Carpobrotus* spp. The results obtained for those habitats are therefore in line with the findings of studies addressing the impact on scrub vegetation [43,45]. Again, however, the research effort focused on sand dune habitat, making the results presented with this study as an update on the knowledge on this topic recognizing the negative impact of *Carpobrotus* spp. on Mediterranean coastal scrubs and garrigues. We note also that the alpha diversity trends registered slightly differ in the two years of sampling. This phenomenon is probably due to the variation in *Carpobrotus* spp. cover in the two years (lower in 2021). Indeed, we recorded a reduction in *Carpobrotus* spp. cover due to the desiccation of some fresh branches (with a correlated increase in dead litter) probably as a consequence of the harsher climatic condition (in particular very low precipitation) to which communities investigated were subjected in the summer of 2020 and spring of 2021 (see Appendix A). In this year, in fact, the particular aridity may have decreased the diversity in the natural control habitats (due to a lack of annual species) and reduced the dominance of *Carpobrotus* spp. (which was less abundant) in invaded ones. In particular, habitat 1430, being more structured (more presence of scrubby species) and generally suffering a slightly less pronounced impact than habitat 5320, recorded a greater coverage of native species in the invaded areas than in 2020, with a correlated increase in community diversity.

Our results demonstrate that the invasion of *Carpobrotus* spp. affects negatively rocky coastal habitats not only from the quantitative point of view (i.e., alpha diversity) but also from the compositional one. *Carpobrotus* spp. dramatically induced a change in the composition of invaded communities, which resulted as shifted (especially communities of habitat 1240 and habitat 5320) from the not-invaded ones, toward smaller sets of species. The presence of many typical species of the sampled habitats as indicator species only in the not-invaded communities indicate their almost complete depletion in the invaded ones of the same habitat typology, testifying the very important level of replacement caused by *Carpobrotus* spp. invasion. This confirms that its successful establishment probably operates through the replacement and exclusion of native species, rather than coexistence [15,20]. Moreover, as shown in the NMDS analyses, it is worth mentioning that the communities invaded by *Carpobrotus* spp. are characterized by the presence of some more ruderal and nitrophilous species (e.g., *Cynodon dactylon* (L.) Pers. and *Mercurialis annua* L.). This effect, even if in our case not so pronounced, is comparable with the one observed by Buisson et al. [43] for rocky habitats and by other authors for sandy ones [22]. Indeed, it has already been demonstrated that *Carpobrotus* spp. may favor the replacement of native plants by ruderal nitrophilous species through soil nutrient enrichment [41,46], hence also in our case the presence of ruderal species could be linked to the significant presence of *Carpobrotus* spp. litter (as shown in Figure 1B). Moreover, despite the three habitats being naturally characterized by completely separated sets of species, *Carpobrotus* spp. induced a strong homogenization of their flora. In fact, it has been frequently reported that serious invaders may directly and quickly lead to a strong biotic homogenization of plant communities [47,48,49,50,51]. The impact of *Carpobrotus* spp. is an example of this phenomenon, as demonstrated by this study and by others on dune habitats [22].

The analysis of beta diversity components of diversity allowed us to confirm the insight from compositional analyses, shedding light on the main ecological processes in action due to the invasion. For habitat 1240, the higher beta diversity between invaded and not-invaded communities is primarily a consequence of the reduced number of species occurring in contexts impacted by *Carpobrotus* spp., confirming the results of alpha diversity. The depletion of invaded communities, in terms of the number of species, is responsible for the high beta diversity between invaded and not invaded communities being due mainly to the turnover of species, with *Carpobrotus* spp. replacing (in many cases entirely) and outcompeting native species. This evidence confirms the already shown phenomenon of a shift in community composition due to *Carpobrotus* spp. (e.g., [20]), even considering the beta diversity. This pattern has been found also for habitat 5320, in which the diversity between invaded and not-invaded communities was considerably higher than that occurring within them. Hence the impact of *Carpobrotus* spp. induces even in garrigues a severe change in plant communities, leading them to be considerably differentiated from the native ones. In this case, we note a significant difference in the contribution of turnover and nestedness to beta diversity for the two years. As already stated for other results, this effect might be attributed to the drier condition of 2021 and to the consequent reduction in *Carpobrotus* spp. coverage and the slightly higher affirmation of a few native species more resistant to aridity (e.g., *Euphorbia seguieriana* Neck. and *Helichrysum litoreum* Guss.). Consequently, in terms of beta diversity, this entails that in 2020, the diversity between invaded and not-invaded communities is almost entirely due to species turnover (*Carpobrotus* spp. replacing a large set of native species), while in 2021, the diversity is partially due to nestedness (as the few native species in invaded communities consist of subsets of the invaded one). Finally, the comparable patterns of beta diversity between and within invaded and not-invaded communities can be explained by the intrinsic heterogeneity of this habitat. Despite at the alpha level the effect of *Carpobrotus* spp. invasion is evident through the reduction in diversity, at the beta diversity level it is less readable, since the diversity is relatively high even within invaded and not-invaded patches. Even in this case the harsher condition of 2021 might have influenced the patterns, which in this case led to a slightly higher beta diversity within invaded communities, probably as a consequence of the less abundance of *Carpobrotus* spp. in favor of a small recolonization by native species. To our knowledge, the impacts of invasive species have been taken into account addressing almost exclusively the alpha component of diversity, with few examples of studies including the beta diversity component [40,52], of which only one addresses *Carpobrotus* spp. invasion in coastal areas [53].

In conclusion, within this study, we verified the deleterious impacts on native plant communities linked to the invasion of *Carpobrotus* spp. on the coastal habitats of a small Mediterranean island (N2000 habitats: 1240 = vegetated sea cliffs of the Mediterranean coasts with endemic *Limonium* spp., 1430 = Halo-nitrophilous scrubs (*Pegano*-*Salsoletea*) and 5320 = Low formations of *Euphorbia* close to cliffs). These impacts spanned from the decrease in species richness, diversity and abundance to a compositional shift in invaded communities, which also emerged through beta diversity, with *Carpobrotus* spp. replacing a large set of native species. Indeed, as already shown in several other cases, our data showed that the replacement and exclusion of native species typical of the natural, rather than coexistence, are the main ecological processes linked to *Carpobrotus* spp. invasion. These results allowed us to outline the current impact of *Carpobrotus* spp. on the plant communities of Giglio Island, constituting an important baseline of data in view of the interventions aiming to control this invasive species foreseen within project LIFE LETSGO GIGLIO. Moreover, we confirmed the impacts of *Carpobrotus* spp. invasion on habitats less frequently mentioned in the literature, such as sea rocky cliffs such as 1240 and 5320, and also more structured habitats such as 1430.

## 4. Materials and Methods

### 4.1. Study Area

This study took place on the Island of Giglio (WGS84: 42.35527° N, 10.90134° E), which, with its 21.2 km^2^, is the second largest island in the Tuscan Archipelago (Tyrrhenian Sea, Italy) and is located about 14 km in front of Monte Argentario promontory in south Tuscany (Figure 5). The island’s territory is predominantly mountainous, with generally very steep slopes and extensive stretches of denuded rock both inland and along the coast. The climate is Mediterranean, with mild, rainy winters and hot, dry summers peaking in July and August, followed by an autumn resumption of rainfall [54].

The vegetation is purely Mediterranean with the presence of more or less recent holm oak woods and various types of thickets and garrigue [54]. The abandonment of traditional agricultural practices and the shift towards a tourist-based economy has led to a major transformation of the vegetation landscape over the last 70 years, with an increase in the number of thickets and scrub encroaching on former crops (although the last decade has seen an increase in the area returned to cultivation), and an increased impact on coastal habitats [44,54].

Giglio Island is almost entirely included in the Isola del Giglio SAC/SPA (IT51A0023), which covers about 21 km^2^, while the area included within the boundaries of the Tuscan Archipelago National Park is smaller (8.9 km^2^). The only three small towns remain outside the SAC/SPA: Giglio Castello, Giglio Porto, and the hamlet of Campese, home to the approximately 1550 inhabitants living on the island.

The rocky cliffs coastal vegetation of the island is of particular interest for this study, being the one invaded by *Carpobrotus* spp. and hosts a mosaic of habitats of conservation interest according to Directive 92/43/EEC “Habitat” including the habitat of vegetated sea cliffs of the Mediterranean coasts with endemic *Limonium* spp. (habitat code 1240 according to Dir. 92/43/EEC, and including the important endemic *Limonium sommerianum* Fiori, see also [55], the habitat of Halo-nitrophilous scrubs (*Pegano*-*Salsoletea*) (habitat code 1430, represented at Giglio Island by the subtypes identified with the alliance *Artemision arborescentis*) and the habitat of Low formations of *Euphorbia* close to cliffs (hab. code 5320). Particularly, monitoring has been carried out in coastal vegetation at two specific sites strongly invaded by *Carpobrotus* spp.: the promontory of “Punta Capel Rosso” (the island’s southern tip, where *Carpobrotus* spp. is invading two main types of habitats: 1240 and 5320) and the promontory of “Punta del Fenaio” (the island’s northern tip, where *Carpobrotus* spp. is invading two main types of habitats: 1240 and 1430).

### 4.2. Data Collection

The survey of the vegetation was performed at two locations: Capel Rosso and Fenaio, and was stratified according to the EU habitat, as mapped according to the HaSCITu (Habitat in the Sites of Conservation Interest in Tuscany) program (http://www.regione.toscana.it/-/la-carta-degli-habitat-nei-siti-natura-2000-toscani, accessed on 16 October 2022). During the first phases of the project (between February and late May 2020), several inspections of the islands allowed very detailed mapping of the distribution of *Carpobrotus* spp. in the study area. The experimental monitoring design involves the floristic survey of 2 × 2 m square plots in invaded and control areas. The plots are permanent and were positioned according to a stratified random design based on the surface area of the habitats affected by the actions. For each invaded plot, a paired control as close as possible was selected. Specifically, 44 permanent plots were placed (12 × 2 plots for Habitat 1240, 6 × 2 plots for Habitat 5320, and 4 × 2 plots for Habitat 1430). The higher number of replies for habitat 1240 is linked to the higher surface occupied by this habitat (and invaded by *Carpobrotus* spp.) in the area of study. Each plot was georeferenced, and vegetation sampling was carried out during the vegetative period (May–June) in 2020 and 2021: these two years represent the baseline of data for long-term monitoring of the future interventions of removal and will be carried out for the following years after the interventions to observe the evolution of the vegetation. In each plot, we collected information on the cover of the fresh *Carpobrotus* spp. mat and of its dead litter, as well as of each native species using a percentage scale, taking into account the overlapping of different species (total cover was recorded). Repeated sampling over 2 years led to the survey of 88 plots.

### 4.3. Statistical Analyses

The effects of the invasion by *Carpobrotus* spp. on alpha diversity of native vegetation, and particularly on the habitats worthy of conservation 1240, 1430, and 5320 were evaluated using Repeated Measurement ANOVA-type models fitting a series of Generalized Least Squares models (GLSm), accounting for a Gaussian spatial correlation of the observations (linked to both the paired structure of sampling design and the presence of two separated localities) and taking also into account that the same plot was revisited for the two years of sampling. For each of the three habitats separately, we assessed whether native species richness (SR), native species diversity expressed as H’ index and native species abundance (expressed as the sum of percentage cover of each species) varied according to the status of invasion (invaded vs. control plots). The response variables were log-transformed, when needed, to achieve normality of residuals.

We studied the changes in the species composition of plots using multivariate analyses, including in the same analysis plots from all three habitat types. Plot species composition differences were analyzed using a non-metric multidimensional scaling (NMDS) analysis based on Bray–Curtis dissimilarities calculated on abundance data (expressed as percentages). We further evaluated the role of particular species in the species turnover due to the invasion process by carrying out an Indicator Species Analysis (ISA, [56]). The ISA allows computing an indicator value d (ranging between 0 and 100) of each species as the product of the relative frequency and relative average abundance of species in clusters. The analysis also produces a significance value, representing the probability of obtaining a d value as high as that observed over 9999 iterations. We conducted the analyses by merging the two years of survey and adopting a grouping based on status and invasion and habitat.

Finally, we further evaluated the beta diversity patterns between invaded and control plots within each habitat type for both years, calculating the distance matrices accounting for spatial turnover, nestedness, and the sum of both components [37], using species presence/absence data. We used Sørensen’s index to quantify dissimilarity.

All analyses were conducted in R environment (R version 4.1.0): the GLS models were fitted using the ‘nlme’ package version 3.1-15 [57]; the NMDS was produced using the ‘vegan’ package version 2.5-7 [58]; the ISA was conducted using the package ‘labdsv’ (R package version 1.8-0, https://CRAN.R-project.org/package=labdsv, accessed on 16 October 2022) and the beta diversity was calculated using the package ‘betapart’ [59]. All plots were drawn using ‘ggplot2′ package version 3.3.3 [60].

## Figures and Tables

**Figure 1 plants-11-02802-f001:**
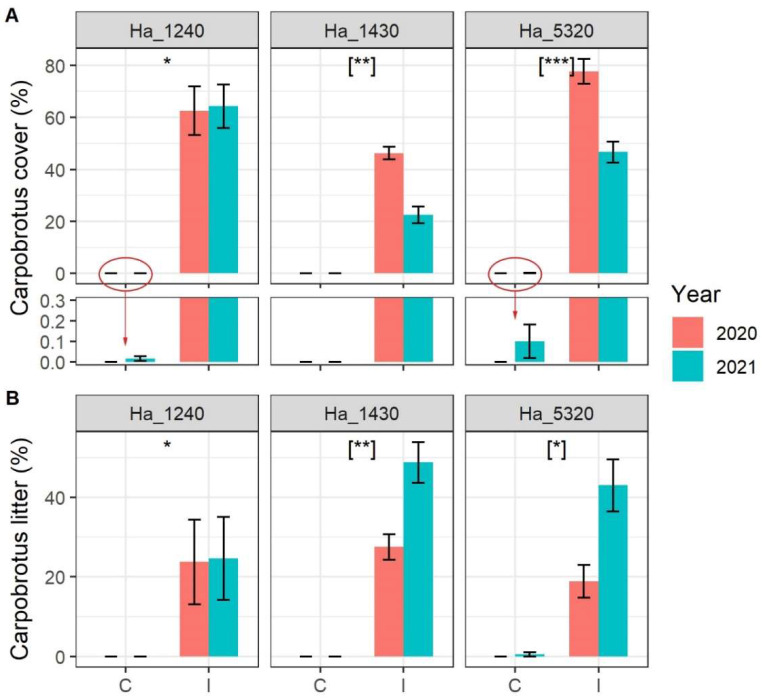
Cover of (**A**) *Carpobrotus* spp. and (**B**) *Carpobrotus* spp. dead litter in the sampled plot according to N2000 habitat, status of invasion (C = Control plots, I = Invaded plots) and year. N2000 habitat codes: 1240 = vegetated sea cliffs of the Mediterranean coasts with endemic *Limonium* spp., 1430 = Halo-nitrophilous scrubs (*Pegano*-*Salsoletea*) and 5320 = Low formations of *Euphorbia* close to cliffs. Error bars correspond to standard error. Asterisks indicate significant differences between control and invaded plots, [when in square bracket indicate significant differences within the interaction term Invasion Status:Year]. Significance codes: *p* value < 0.001 ‘***’; *p* value < 0.01 ‘**’; *p* value < 0.05 ‘*’. In lower panel of (**A**), the *Carpobrotus* cover scale is highly magnified to allow the reading of very small values of cover.

**Figure 2 plants-11-02802-f002:**
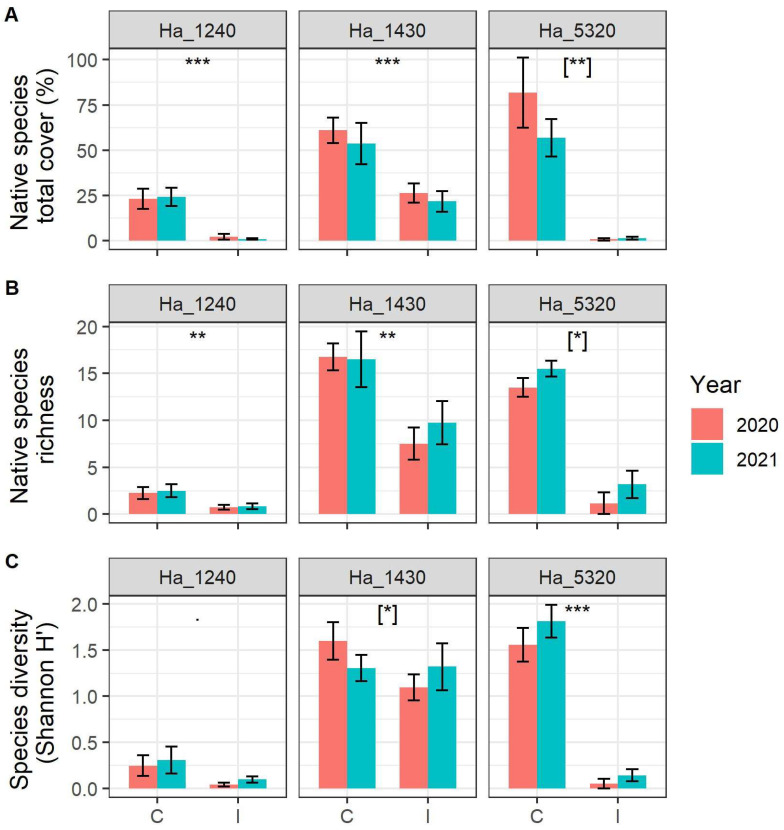
(**A**) Native species richness, (**B**) native species diversity expressed as Shannon index H’ and (**C**) native species abundance (expressed as sum of percentage cover of each species) in the sampled plot according to N2000 habitat, status of invasion (C = Control plots, I = Invaded plots) and year. N2000 habitat codes: 1240 = vegetated sea cliffs of the Mediterranean coasts with endemic *Limonium* spp., 1430 = Halo-nitrophilous scrubs (*Pegano*-*Salsoletea*) and 5320 = Low formations of *Euphorbia* close to cliffs. Error bars correspond to standard error. Asterisks indicate significant differences between control and invaded plots, [when in square bracket indicate significant differences within the interaction term Invasion Status:Year]. Significance codes: *p* value < 0.001 ‘***’; *p* value < 0.01 ‘**’; *p* value < 0.05 ‘*’, *p* value < 0.10 ‘˙’.

**Figure 3 plants-11-02802-f003:**
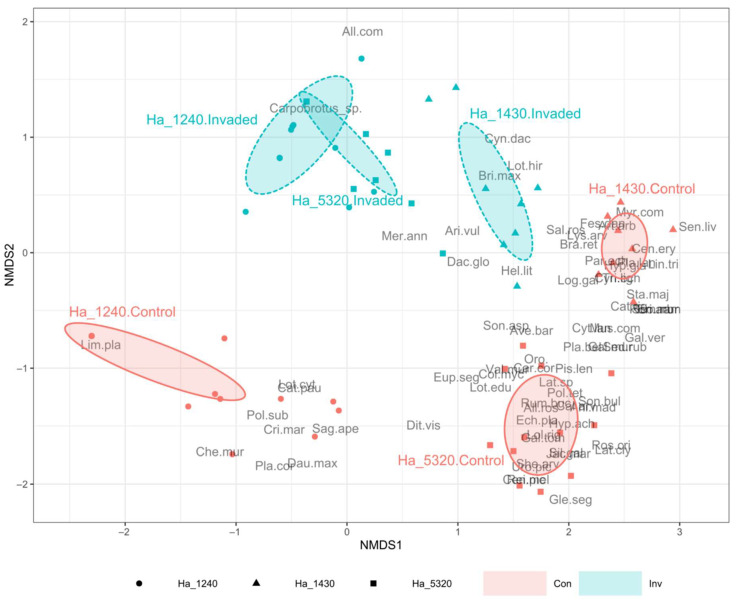
Non-metric multidimensional scaling (NMDS) ordination plot based on Bray–Curtis dissimilarities of the 88 sampled plots. Plots are grouped according to N2000 habitat and invasion status (Control plots vs. Invaded plots). Ellipses represent the standard deviation of sampled plot positions. N2000 habitat codes: Ha_1240 (circles) = vegetated sea cliffs of the Mediterranean coasts with endemic *Limonium* spp.; Ha_1430 (triangles) = Halo-nitrophilous scrubs (*Pegano*-*Salsoletea*) and Ha_5320 (squares) = Low formations of *Euphorbia* close to cliffs. See Appendix A for plant full names. Invasion Status codes: Con (red)= Control plots; Inv (blue) = Invaded plots.

**Figure 4 plants-11-02802-f004:**
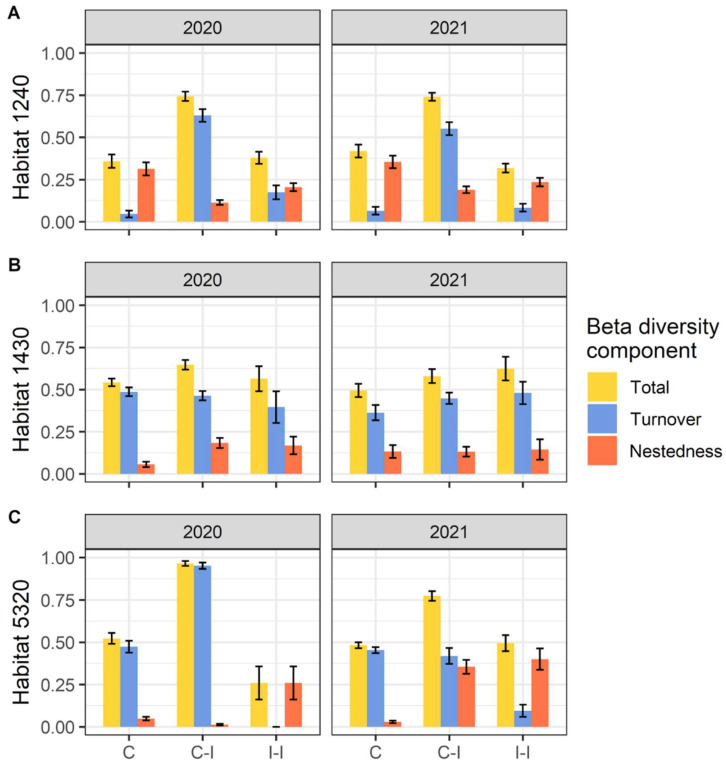
Total beta diversity and Turnover and Nestedness components in (**A**) habitat 1240 = vegetated sea cliffs of the Mediterranean coasts with endemic *Limonium* spp., (**B**) habitat 1430 = Halo-nitrophilous scrubs (*Pegano*-*Salsoletea*) and (**C**) habitat 5320 = Low formations of *Euphorbia* close to cliffs. Beta diversity components are separated per pairwise comparison (C = Control plots, I = Invaded plots) and year. Error bars correspond to standard error.

**Figure 5 plants-11-02802-f005:**
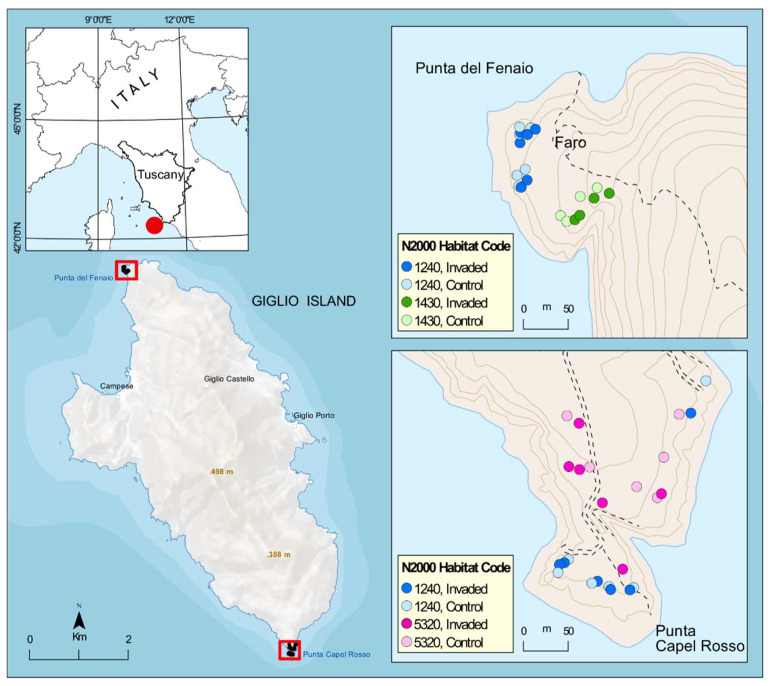
Area of study and distribution of the 44 sampling plots according to N2000 habitat and status of invasion. N2000 habitat codes: 1240 = vegetated sea cliffs of the Mediterranean coasts with endemic *Limonium* spp., 1430 = Halo-nitrophilous scrubs (*Pegano*-*Salsoletea*) and 5320 = Low formations of *Euphorbia* close to cliffs.

**Table 1 plants-11-02802-t001:** Repeated Measurement ANOVA table for the effect of Invasion Status (Control plots vs. Invaded plots) and Year (sampling year 2020 vs. 2021) on Native species total cover, native species richness and native species diversity expressed as H’, provided for each Natura 2000 habitat. N2000 habitat codes: 1240 = vegetated sea cliffs of the Mediterranean coasts with endemic *Limonium* spp., 1430 = Halo-nitrophilous scrubs (*Pegano-Salsoletea*) and 5320 = Low formations of *Euphorbia* close to cliffs. numDF: numerator degree of freedom; denDF: denominator degree of freedom. Significance codes: *p* value < 0.001 ‘***’; *p* value < 0.01 ‘**’; *p* value < 0.05 ‘*’, *p* value < 0.10 ‘˙’.

Response Variable	Habitat	Variable	numDF	denDF	F Value	*p* Value	
Native species total cover	1240 ^£^	Invasion Status	1	44	155.35	<0.001	***
		Year	1	44	0.25	0.616	
		Invasion Status:Year	1	44	0.54	0.466	
	1430 ^£^	Invasion Status	1	12	19.69	<0.001	***
		Year	1	12	4.46	0.056	˙
		Invasion Status:Year	1	12	0.15	0.705	
	5320 ^£^	Invasion Status	1	20	101.44	<0.001	***
		Year	1	20	0.13	0.720	
		Invasion Status:Year	1	20	9.21	0.007	**
Native species richness	1240 ^£^	Invasion Status	1	44	8.32	0.006	**
		Year	1	44	0.45	0.504	
		Invasion Status:Year	1	44	0.11	0.743	
	1430	Invasion Status	1	12	18.24	0.001	**
		Year	1	12	0.51	0.487	
		Invasion Status:Year	1	12	0.80	0.388	
	5320 ^£^	Invasion Status	1	20	35.09	<0.001	***
		Year	1	20	15.75	<0.001	***
		Invasion Status:Year	1	20	8.12	0.010	*
Species diversity (H’)	1240 ^£^	Invasion Status	1	44	3.93	0.054	˙
		Year	1	44	4.80	0.034	*
		Invasion Status:Year	1	44	0.29	0.595	
	1430	Invasion Status	1	12	3.46	0.088	˙
		Year	1	12	0.15	0.703	
		Invasion Status:Year	1	12	9.29	0.010	*
	5320	Invasion Status	1	20	93.61	<0.001	***
		Year	1	20	3.23	0.087	˙
		Invasion Status:Year	1	20	0.74	0.400	

^£^ Variables log transformed.

**Table 2 plants-11-02802-t002:** Results of the Indicator Species Analysis according to N2000 habitat and invasion status (Control plots vs. Invaded plots). N2000 habitat codes: 1240 = vegetated sea cliffs of the Mediterranean coasts with endemic *Limonium* spp., 1430 = Halo-nitrophilous scrubs (*Pegano*-*Salsoletea*) and 5320 = Low formations of *Euphorbia* close to cliffs. Significance codes: *p* value < 0.001 ‘***’; *p* value < 0.01 ‘**’; *p* value < 0.05 ‘*’.

Species	Invasion Status	Habitat	Indicator Value	*p* Value	
*Limonium planesiae* Pignatti	Control plots	1240	0.95	<0.001	***
*Brachypodium retusum* (Pers.) P.Beauv.		1430	0.70	<0.001	***
*Artemisia arborescens* (Vaill.) L.			0.69	<0.001	***
*Lysimachia arvensis* (L.) U.Manns & Anderb.			0.64	<0.001	***
*Helichrysum litoreum* Guss.			0.52	<0.001	***
*Paronychia echinulata* Chater			0.50	<0.001	***
*Linum trigynum* L.			0.38	0.001	**
*Festuca danthonii* Asch. & Graebn. subsp. *danthonii*			0.34	0.001	**
*Lotus hirsutus* L.			0.34	0.003	**
*Hypochaeris glabra* L.			0.30	0.007	**
*Arisarum vulgare* O.Targ.Tozz. subsp. *vulgare*			0.29	0.015	*
*Myrtus communis* L.			0.25	0.014	*
*Plantago lanceolata* L.			0.25	0.015	*
*Stachys major* (L.) Bartolucci & Peruzzi			0.25	0.015	*
*Sedum rubens* L.			0.23	0.029	*
*Muscari comosum* (L.) Mill.			0.22	0.038	*
*Cytisus laniger* DC.			0.20	0.026	*
*Allium roseum* L. subsp. *roseum*		5320	0.89	<0.001	***
*Euphorbia segetalis* L.			0.80	<0.001	***
*Lotus edulis* L.			0.75	<0.001	***
*Polycarpon tetraphyllum* (L.) L.			0.69	<0.001	***
*Hypochaeris achyrophorus* L.			0.57	<0.001	***
*Carlina corymbosa* L.			0.57	<0.001	***
*Glebionis segetum* (L.) Fourr.			0.50	<0.001	***
*Urospermum picroides* (L.) Scop. ex F.W.Schmidt			0.50	<0.001	***
*Rumex bucephalophorus* L.			0.49	<0.001	***
*Coleostephus myconis* (L.) Cass. ex Rchb.f.			0.48	<0.001	***
*Valantia muralis* L.			0.42	<0.001	***
*Lolium rigidum* Gaudin			0.40	<0.001	***
*Jacobaea maritima* (L.) Pelser & Meijden subsp. *maritima*			0.39	<0.001	***
*Silene gallica* L.			0.33	0.002	**
*Echium plantagineum* L.			0.32	0.006	**
*Avena barbata* Pott ex Link			0.31	0.022	*
*Pistacia lentiscus* L.			0.28	0.019	*
*Anisantha madritensis* (L.) Nevski subsp. *madritensis*			0.27	0.018	*
*Rostraria cristata* (L.) Tzvelev			0.25	0.024	*
*Plantago bellardii* All.			0.23	0.028	*
*Calendula arvensis* (Vaill.) L.			0.17	0.049	*
*Dactylis glomerata* L.	Invaded plots	1430	0.59	<0.001	***
*Briza maxima* L.			0.34	0.005	**
*Cynodon dactylon* (L.) Pers.			0.26	0.008	**
*Carpobrotus* spp.		5320	0.39	<0.001	***

## Data Availability

Data are contained within the article or Appendix A.

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
