# Peer review of "Impacts of the Invasive Alien Carpobrotus spp. on Coastal Habitats on a Mediterranean Island (Giglio Island, Central Italy)"

_plants, 2022, doi:10.3390/plants11202802_

Round 1
Reviewer 1 Report
Dears,
very interesting paper, particularly due to comparing different habitats.
I have only few remarks.

Author Response
Dear Reviewer,
We thank for the comments made on our manuscript.
You can find below a point by pont list of the responses to your comments and suggestions.
36, 40: please unify writing of IAPS
R: Done.
67, 97 LIFE
R: Done.
104 … of invaded and native communities. Yes, I know what you mean, but both communities are native to Italy. So, in my opinion, the term should be “invaded and notinvade communities”
R: Done.
Table 1: reSponse variable and variable please left aligned
R: Done.
Why you use Shannon and native species richness? If the plot size is the same, richness is enough. If you want to use Shannon as eveness measure, please explain, why the results are different to richness.
R: Shannon diversity and native species richness are different measures of Alpha diversity, which are often used in a complementary way to describe different features of the communities diversity (Total number of species and Evenness of communities). Differences are already discussed in the discussion.
Table 1: is it explained, what is invasion status and what is year?
R: We added the specification in the captions.
120: Why there is a decrease in Carpobrotus cover between 2020 and 2021? – 221: harsher climate conditions? But Carpobrotus in not annual species.
R: This phenomenon is probably a consequence of the harsher climatic condition (in particular in higher aridity) to which communities investigated were subjected in 2021. Indeed, even if it is not an annual species we recorded a reduction of Carpobrotus cover due to the desiccation of some fresh branches (with a correlated increase in dead litter) we added these considerations in the results and in the discussion.
Fig 1A unclear, line 129,130: if C is the control, it is not possible that the “zoomed” are the seedlings. Otherwise in 1A please add: J (for Juvenile) for each of the two left bars, and A for Adult for each of the two right bars. And in 1B: C, and I for control and invaded. But what is in this case control and invaded in Fig. 1A. Sorry unclear!
R: The plots are zoomed to allow the reading of very small cover values linked to the arrival of Carpobrotus seedlings in control plots from 2020 to 2021, as described in lines XX-XX. We added a specification stating that “In lower panel of A), the Carpobrotus cover scale is highly magnified to allow the reading of very small values of cover.”
Table 2: Species left aligned, please. Add a line between control and invaded plots, so that it is more clear.
R: Done.
Is it possible to add a frequency table, so that it is clear, how Carpobrotus decrease the native species. Table for 1430 and 5320 and 1240, control/invaded plots.
R: Other than in Figure 2 species per plot data are provided as Supplementary Table S2.
In general, considering the ecosystem: is there a change in total vegetation cover of the plots, between invaded and not-invaded plots? Is there less bare ground? And what is vegetation structure? Unified?
R: We added a few lines at the beginning of the Results section and in the Supplementary Table S3.
241: why there are in invaded plots more nitrophilous species? Is Carpobrotus able to fix aerial-nitrogen? Or is it, because Carpobrotus establishes easier at ruderal, and thus nitrophilous stands?
R: As already demonstrated by Fried et al. 2014 (but see also Campoy et al. 2018), Carpobrotus is responsible for a shifting in species composition of invaded communities toward more nitrophilous species. As for other invasive species, this is due to changes in the upper soil profile attributed to the abundant litter produced.
Fried, G., Laitung, B., Pierre, C. et al. Impact of invasive plants in Mediterranean habitats: disentangling the effects of characteristics of invaders and recipient communities. Biol Invasions 16, 1639–1658 (2014). https://doi.org/10.1007/s10530-013-0597-6
245: How is it possible that Carpobrotus enrich soil nutrients?
R: As explained in the reply to previous comment, the abundant production of litter by Caropobrotus alters the upper soil profile balance in nutrients, favouring the establishing of more nitrophilous species (see Campoy et al. 2018 and Fried et al. 2014).
Reviewer 2 Report
The authors of this study examined the impact of the invasive species Carpobrotus edulis and C. acinaciformis on the biodiversity of several natural habitats on the island of Giglio (Italy). The experimental design fits the objectives of the study, the results are clearly presented and the discussion includes solid arguments. The experiment includes several natural habitats and a high number of replicates, which is positive. I find it a bit odd that the number of replicates differs between habitats (24 plots for Habitat 1240 versus 8 plots for Habitat 1430), but that is not a major concern. The fact that the authors repeated the measurements in two different years adds value to the results obtained. I believe that this work is an original and valuable contribution, and therefore deserves to be published.
Minor suggestions:
L276 “Despite at alpha level” -> “Despite that at the alpha level”
Author Response
Dear Reviewer,
We thank for the comments made on our manuscript.
You can find below a point by pont list of the responses to your comments and suggestions.
The authors of this study examined the impact of the invasive species Carpobrotus edulis and C. acinaciformis on the biodiversity of several natural habitats on the island of Giglio (Italy). The experimental design fits the objectives of the study, the results are clearly presented and the discussion includes solid arguments. The experiment includes several natural habitats and a high number of replicates, which is positive. I find it a bit odd that the number of replicates differs between habitats (24 plots for Habitat 1240 versus 8 plots for Habitat 1430), but that is not a major concern. The fact that the authors repeated the measurements in two different years adds value to the results obtained. I believe that this work is an original and valuable contribution, and therefore deserves to be published.
R: We thank the reviewer for his/her consideration of our work.
Minor suggestions:
L276 “Despite at alpha level” -> “Despite that at the alpha level”
R: Done.
Reviewer 3 Report
The manuscript entitled “Impacts of the invasive alien Carpobrotus spp. on coastal habitats in a Mediterranean island (Gioglio island, Central Italy)” exposes the results after a two year evaluation on the community impacts of an important invasive species in Mediterranean habitats prior to the start of a management project. The invasive plant Carpobrotus spp (C. edulis, C. acinaciformis, and hybrids) are considered one of the worst invasive species in Europe, and particularly of interest for conservation of the Mediterranean biodiversity hotspot, due to the success this species has shown in this area. Although the impacts of this invader into community richness are well-known, the novelty of this study resides in the multihabitat approach and the results regarding the species turnover after Carpobrotus spp. invasion. The authors collected data two years during vegetative period regarding species composition in 4m2 invaded and non-invaded fixed plots from three different Nature 2000 habitats (n = 4-12). This is the first time the impacts of this species in community composition in sea cliff communities are reported. Those habitats are of special conservation value.
The manuscript is very well-written and the collected data and discussion presented are very interesting and contribute to a better understanding of this invasive species impacts. However, I have some concerns about the methodology description and some potential improvements for the different figures. In some cases, more methodological detail is required to guide readers of the journal in issues they may not be familiar with. Moreover, I have some suggestions on data analysis strategies that should be explored in case they have a significant effect on the exposed results.
My major concern is that the authors haven’t statistically explored the role of Carpobrotus spp. abundance role on the community composition impact. The authors should explore if a higher presence of the invader in the plot leads to significantly different changes into the invaded community. Particularly as the authors saw that the variation in Carpobortus spp cover in the two years (lower in 2021) may be responsible of differences between years, and also as some Carpobortus spp seedlings appear in non-invaded plots. There are different strategies that can be used to address this, such as adding invasive coverage as a model covariate, or explore correlations between diversity, native species abundance and richness and Carpoborotus abundance.
Besides this major point, I will outline here the different issues/questions I have encountered that should be addressed prior publication:
Abstract:
- L20: 4m2 instead of 4m2
- The reader may not be familiar with alpha and beta diversity. Provide a small description for both here.
Introduction:
- L41: Introduce Mediterranean biodiversity hotspot. The impact of the invader is more concerning in those areas because they are particularly hosting large numbers of native species.
- L73: Please detail what is the value of the habitats you have studied. Do they host endangered species? Are they abundant? Do they provide key ecosystem services?
- L82-L86: Please rephrase, and add a description of Beta-diversity
- L90: “beta diversity” instead of beta-diversity. Check consistency across the manuscript. Also sometimes the authors use different designations that may be confusing (alpha diversity, species diversity, H’)
- L90: use italics for “e.g.” across the manuscript.
Results:
- L134 and L150: As you are exposing the results from a figure but also discussing their statistical significance please add also citation to Table 1 as follows: (Figure X, Table 1).
Figures:
- Please report in figures 1,2,4 if the error bars correspond to standard deviation or standard error, and report also the replicate numbers for each habitat.
- I would suggest the authors to rethink colours. For example, It could help the reader to understand easily figure 1 if panel A has a colour palette of vivid colours and panel B has a darker palette, symbolising the Carpobrotus spp. cover and dead litter cover respectively. Also the current focus of the colour is between years, but the main important results are between treatments. Consider using different palettes for invaded and control plots values.
- The authors presented the statistics in Table 1. I appreciate that they have reported all statistical results but I would suggest that they consider including somehow this statistical results into the figures, so the reader can see the mean values and the significance between them. Consider for instance adding asterisks in Figure 1,2 and 4 when differences between control and invaded treatments occurred.
- I suggest to reorder the panels in Figure 2 as the order in Table 1 and the use of the same variables name: “native species total cover”, “Species diversity” and “Native species richness”.
- Figure 3: Please clarify what the ellipses represent. Do they represent confidence intervals? Enlarge also the size font in this figure. It is difficult to interpret species positions in the NMDS. Consider plotting only those species that mainly contribute to global variation (for example >0.5 correlation coefficient). Add colour to the points considering if they are invaded or control.
- Figure 4: I suggest the author collapse the Total, Turnover and Nestedness into a single column, building stacked bargraphs with error bars. This will facilitate reader interpretation of the amount of proportion explained by each element, and will result in a more effective figure.
Discussion:
- L205-208 Add reference
- L208-210 Add reference
- L220 The role of the climate is discussed here for the first time. Please provide data supporting this in supplementary material
- I suggest the authors include discussion on the interaction of the invasive species with key highly valuable species in terms of conservation purposes (such as Limonium sp.).
- L245: I suggest the authors link here the association of Carpobrotus spp. with ruderal and nitrophilous species with their results of percentage of death litter (Figure 1B) that this species left.
- L267, L275: I suggest not using “beta” alone but using always the whole term “beta diversity” instead.
Material and methods
- L343: Please add “was performed at two regions”, as follows: “The survey of the vegetation was performed at two regions: Capel Rosso and Fenaio, and was stratified….”
- L349: Please provide detail on how the controls were selected. How did the authors ensure it was an uninvaded area? Provide also detail on how much Carpobrotus sp. was present in invaded plots. Was 100% invasive coverage excluded?
- Considering the plot distribution, it seems that the control and invaded plots paired. How was this taken into account at the statistical model? Did you provide some control for spatial autocorrelation?
- L395: Does overlap was considered when accounting for species coverage? Provide details if the authors measured total or relative coverage.
- Alpha and beta diversity are not explained enough in this section. Consider adding an explanation for these terms.
- Why was the Indicator Species Analysis (ISA) used and not other similar tools such as SIMPER analysis?
Author Response
Dear Reviewer,
We thank for the comments made on our manuscript.
You can find attached a file including the responses to your comments and suggestions.
